# Joint distribution optimal transportation for domain adaptation

**Nicolas Courty**[*]
Université de Bretagne Sud,
IRISA, UMR 6074, CNRS,
courty@univ-ubs.fr

**Rémi Flamary**[*]
Université Côte d'Azur,
Lagrange, UMR 7293 , CNRS, OCA
remi.flamary@unice.fr

**Amaury Habrard**
Univ Lyon, UJM-Saint-Etienne, CNRS,
Lab. Hubert Curien UMR 5516, F-42023
amaury.habrard@univ-st-etienne.fr

**Alain Rakotomamonjy**
Normandie Universite
Université de Rouen, LITIS EA 4108
alain.rakoto@insa-rouen.fr

## Abstract

This paper deals with the unsupervised domain adaptation problem, where one wants to estimate a prediction function $f$ in a given target domain without any labeled sample by exploiting the knowledge available from a source domain where labels are known. Our work makes the following assumption: there exists a non-linear transformation between the joint feature/label space distributions of the two domain $\mathcal{P}_s$ and $\mathcal{P}_t$ that can be estimated with optimal transport. We propose a solution of this problem that allows to recover an estimated target $\mathcal{P}_t^f = (X, f(X))$ by optimizing simultaneously the optimal coupling and $f$. We show that our method corresponds to the minimization of a bound on the target error, and provide an efficient algorithmic solution, for which convergence is proved. The versatility of our approach, both in terms of class of hypothesis or loss functions is demonstrated with real world classification and regression problems, for which we reach or surpass state-of-the-art results.

## 1 Introduction

In the context of supervised learning, one generally assumes that the test data is a realization of the same process that generated the learning set. Yet, in many practical applications it is often not the case, since several factors can slightly alter this process. The particular case of visual adaptation [1] in computer vision is a good example: given a new dataset of images without any label, one may want to exploit a different annotated dataset, provided that they share sufficient common information and labels. However, the generating process can be different in several aspects, such as the conditions and devices used for acquisition, different pre-processing, different compressions, etc. Domain adaptation techniques aim at alleviating this issue by transferring knowledge between domains [2]. We propose in this paper a principled and theoretically founded way of tackling this problem.

The domain adaptation (DA) problem is not new and has received a lot of attention during the past ten years. State-of-the-art methods are mainly differing by the assumptions made over the change in data distributions. In the *covariate shift* assumption, the differences between the domains are characterized by a change in the feature distributions $\mathcal{P}(X)$, while the conditional distributions $\mathcal{P}(Y|X)$ remain unchanged ($X$ and $Y$ being respectively the instance and label spaces). Importance re-weighting can be used to learn a new classifier (e.g. [3]), provided that the overlapping of the distributions is large

---

[*]Both authors contributed equally.

enough. Kernel alignment [4] has also been considered for the same purpose. Other types of method, denoted as *Invariant Components* by Gong and co-authors [5], are looking for a transformation $\mathcal{T}$ such that the new representations of input data are matching, *i.e.* $\mathcal{P}_s(\mathcal{T}(X)) = \mathcal{P}_t(\mathcal{T}(X))$. Methods are then differing by: *i)* The considered class of transformation, that are generally defined as projections (e.g. [6, 7, 8, 9, 5]), affine transform [4] or non-linear transformation as expressed by neural networks [10, 11] *ii)* The types of divergences used to compare $\mathcal{P}_s(\mathcal{T}(X))$ and $\mathcal{P}_t(\mathcal{T}(X))$, such as Kullback Leibler [12] or *Maximum Mean Discrepancy* [9, 5]. Those divergences usually require that the distributions share a common support to be defined. A particular case is found in the use of optimal transport, introduced for domain adaptation by [13, 14]. $\mathcal{T}$ is then defined to be a push-forward operator such that $\mathcal{P}_s(X) = \mathcal{P}_t(\mathcal{T}(X))$ and that minimizes a global transportation effort or cost between distributions. The associated divergence is the so-called Wasserstein metric, that has a natural Lagrangian formulation and avoids the estimation of continuous distribution by means of kernel. As such, it also alleviates the need for a shared support.

The methods discussed above implicitly assume that the conditional distributions are unchanged by $\mathcal{T}$, *i.e.* $\mathcal{P}_s(Y|\mathcal{T}(X)) \approx \mathcal{P}_t(Y|\mathcal{T}(X))$ but there is no clear reason for this assumption to hold. A more general approach is to adapt both marginal feature and conditional distributions by minimizing a global divergence between them. However, this task is usually hard since no label is available in the target domain and therefore no empirical version $\mathcal{P}_t(Y|X)$ can be used. This was achieved by restricting to specific class of transformation such as projection [9, 5].

**Contributions and outline.** In this work we propose a novel framework for unsupervised domain adaptation between joint distributions. We propose to find a function $f$ that predicts an output value given an input $\mathbf{x} \in \mathcal{X}$, and that minimizes the optimal transport loss between the joint source distribution $\mathcal{P}_s$ and an estimated target joint distribution $\mathcal{P}_t^f = (X, f(X))$ depending on $f$ (detailed in Section 2). The method is denoted as JDOT for "Joint Distribution Optimal Transport" in the remainder. We show that the resulting optimization problem stands for a minimization of a bound on the target error of $f$ (Section 3) and propose an efficient algorithm to solve it (Section 4). Our approach is very general and does not require to learn explicitly a transformation, as it directly solves for the best function. We show that it can handle both regression and classification problems with a large class of functions $f$ including kernel machines and neural networks. We finally provide several numerical experiments on real regression and classification problems that show the performances of JDOT over the state-of-the-art (Section 5).

## 2   Joint distribution Optimal Transport

Let $\Omega \in \mathbb{R}^d$ be a compact input measurable space of dimension $d$ and $\mathcal{C}$ the set of labels. $\mathcal{P}(\Omega)$ denotes the set of all the probability measures over $\Omega$. The standard learning paradigm assumes classically the existence of a set of data $\mathbf{X}_s = \{\mathbf{x}_i^s\}_{i=1}^{N_s}$ associated with a set of class label information $\mathbf{Y}_s = \{\mathbf{y}_i^s\}_{i=1}^{N_s}$, $\mathbf{y}_i^s \in \mathcal{C}$ (the learning set), and a data set with unknown labels $\mathbf{X}_t = \{\mathbf{x}_i^t\}_{i=1}^{N_t}$ (the testing set). In order to determine the set of labels $\mathbf{Y}_t$ associated with $\mathbf{X}_t$, one usually relies on an empirical estimate of the joint probability distribution $\mathcal{P}(X, Y) \in \mathcal{P}(\Omega \times \mathcal{C})$ from $(\mathbf{X}_s, \mathbf{Y}_s)$, and the assumption that $\mathbf{X}_s$ and $\mathbf{X}_t$ are drawn from the same distribution $\mu \in \mathcal{P}(\Omega)$. In the considered adaptation problem, one assumes the existence of two distinct joint probability distributions $\mathcal{P}_s(X, Y)$ and $\mathcal{P}_t(X, Y)$ which correspond respectively to two different *source* and *target* domains. We will write $\mu_s$ and $\mu_t$ their respective marginal distributions over $X$.

### 2.1   Optimal transport in domain adaptation

The Monge problem is seeking for a map $\mathcal{T}_0 : \Omega \to \Omega$ that pushes $\mu_s$ toward $\mu_t$ defined as:

$$\mathcal{T}_0 = \underset{\mathcal{T}}{\operatorname{argmin}} \int_\Omega \mathrm{d}(\mathbf{x}, \mathcal{T}(\mathbf{x})) d\mu_s(\mathbf{x}), \quad \text{s.t. } \mathcal{T} \# \mu_s = \mu_t,$$

where $\mathcal{T} \# \mu_s$ the *image measure* of $\mu_s$ by $\mathcal{T}$, verifying:

$$\mathcal{T} \# \mu_s(A) = \mu_t(\mathcal{T}^{-1}(A)), \quad \forall \text{ Borel subset } A \subset \Omega, \tag{1}$$

and $\mathrm{d} : \Omega \times \Omega \to \mathbb{R}^+$ is a metric. In the remainder, we will always consider without further notification the case where d is the squared Euclidean metric. When $\mathcal{T}_0$ exists, it is called an optimal transport map, but it is not always the case (*e.g.* assume that $\mu_s$ is defined by one Dirac measure and

$\mu_t$ by two). A relaxed version of this problem has been proposed by Kantorovitch [15], who rather seeks for a transport plan (or equivalently a joint probability distribution) $\boldsymbol{\gamma} \in \mathcal{P}(\Omega \times \Omega)$ such that:

$$\boldsymbol{\gamma}_0 = \underset{\boldsymbol{\gamma} \in \Pi(\mu_s, \mu_t)}{\operatorname{argmin}} \int_{\Omega \times \Omega} d(\mathbf{x}_1, \mathbf{x}_2) d\boldsymbol{\gamma}(\mathbf{x}_1, \mathbf{x}_2), \tag{2}$$

where $\Pi(\mu_s, \mu_t) = \{\boldsymbol{\gamma} \in \mathcal{P}(\Omega \times \Omega) | p^+ \# \boldsymbol{\gamma} = \mu_s, p^- \# \boldsymbol{\gamma} = \mu_t\}$ and $p^+$ and $p^-$ denotes the two marginal projections of $\Omega \times \Omega$ to $\Omega$. Minimizers of this problem are called optimal transport plans. Should $\boldsymbol{\gamma}_0$ be of the form $(id \times \mathcal{T}) \# \mu_s$, then the solution to Kantorovich and Monge problems coincide. As such the Kantorovich relaxation can be seen as a generalization of the Monge problem, with less constraints on the existence and uniqueness of solutions [16].

Optimal transport has been used in DA as a principled way to bring the source and target distribution closer [13, 14, 17], by seeking for a transport plan between the empirical distributions of $\mathbf{X}_s$ and $\mathbf{X}_t$ and interpolating $\mathbf{X}_s$ thanks to a barycentric mapping [14], or by estimating a mapping which is not the solution of Monge problem but allows to map unseen samples [17]. Moreover, they show that better constraining the structure of $\boldsymbol{\gamma}$ through entropic or classwise regularization terms helps in achieving better empirical results.

## 2.2  Joint distribution optimal transport loss

The main idea of this work is is to handle a change in both marginal and conditional distributions. As such, we are looking for a transformation $\mathcal{T}$ that will align directly the joint distributions $\mathcal{P}_s$ and $\mathcal{P}_t$. Following the Kantovorich formulation of (2), $\mathcal{T}$ will be implicitly expressed through a coupling between both joint distributions as:

$$\boldsymbol{\gamma}_0 = \underset{\boldsymbol{\gamma} \in \Pi(\mathcal{P}_s, \mathcal{P}_t)}{\operatorname{argmin}} \int_{(\Omega \times \mathcal{C})^2} \mathcal{D}(\mathbf{x}_1, y_1; \mathbf{x}_2, y_2) d\boldsymbol{\gamma}(\mathbf{x}_1, y_1; \mathbf{x}_2, y_2), \tag{3}$$

where $\mathcal{D}(\mathbf{x}_1, y_1; \mathbf{x}_2, y_2) = \alpha d(\mathbf{x}_1, \mathbf{x}_2) + \mathcal{L}(y_1, y_2)$ is a joint cost measure combining both the distances between the samples and a loss function $\mathcal{L}$ measuring the discrepancy between $y_1$ and $y_2$. While this joint cost is specific (separable), we leave for future work the analysis of generic joint cost function. Putting it in words, matching close source and target samples with similar labels costs few. $\alpha$ is a positive parameter which balances the metric in the feature space and the loss. As such, when $\alpha \to +\infty$, this cost is dominated by the metric in the input feature space, and the solution of the coupling problem is the same as in [14]. It can be shown that a minimizer to (3) always exists and is unique provided that $\mathcal{D}(\cdot)$ is lower semi-continuous (see [18], Theorem 4.1), which is the case when $d(\cdot)$ is a norm and for every usual loss functions [19].

In the unsupervised DA problem, one does not have access to labels in the target domain, and as such it is not possible to find the optimal coupling. Since our goal is to find a function on the target domain $f : \Omega \to \mathcal{C}$, we suggest to replace $y_2$ by a proxy $f(\mathbf{x}_2)$. This leads to the definition of the following joint distribution that uses a given function $f$ as a proxy for $y$:

$$\mathcal{P}_t^f = (\mathbf{x}, f(\mathbf{x}))_{\mathbf{x} \sim \mu_t} \tag{4}$$

In practice we consider empirical versions of $\mathcal{P}_s$ and $\mathcal{P}_t^f$, i.e. $\hat{\mathcal{P}}_s = \frac{1}{N_s} \sum_{i=1}^{N_s} \delta_{\mathbf{x}_i^s, \mathbf{y}_i^s}$ and $\hat{\mathcal{P}}_t^f = \frac{1}{N_t} \sum_{i=1}^{N_t} \delta_{\mathbf{x}_i^t, f(\mathbf{x}_i^t)}$. $\boldsymbol{\gamma}$ is then a matrix which belongs to $\Delta$ , i.e.the transportation polytope of non-negative matrices between uniform distributions. Since our goal is to estimate a prediction $f$ on the target domain, we propose to find the one that produces predictions that match optimally source labels to the aligned target instances in the transport plan. For this purpose, we propose to solve the following problem for JDOT:

$$\min_{f, \boldsymbol{\gamma} \in \Delta} \sum_{ij} \mathcal{D}(\mathbf{x}_i^s, \mathbf{y}_i^s; \mathbf{x}_j^t, f(\mathbf{x}_j^t)) \boldsymbol{\gamma}_{ij} \quad \equiv \quad \min_f W_1(\hat{\mathcal{P}}_s, \hat{\mathcal{P}}_t^f) \tag{5}$$

where $W_1$ is the 1-Wasserstein distance for the loss $\mathcal{D}(\mathbf{x}_1, y_1; \mathbf{x}_2, y_2) = \alpha d(\mathbf{x}_1, \mathbf{x}_2) + \mathcal{L}(y_1, y_2)$. We will make clear in the next section that the function $f$ we retrieve is theoretically sound with respect to the target error. Note that in practice we add a regularization term for function $f$ in order to avoid overfitting as discussed in Section 4. An illustration of JDOT for a regression problem is given in Figure 1. In this figure, we have very different joint and marginal distributions but we want

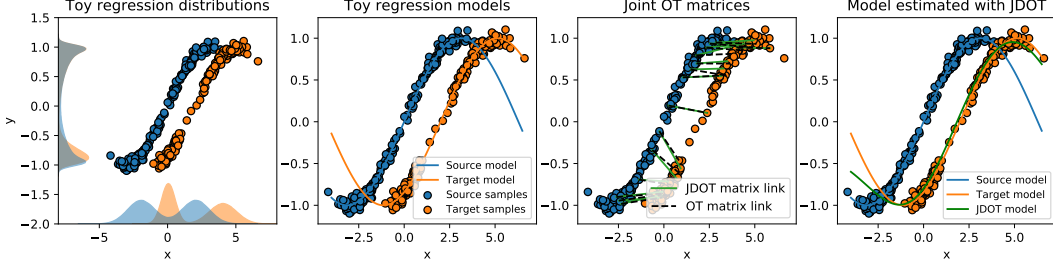

Figure 1: Illustration of JDOT on a 1D regression problem. (left) Source and target empirical distributions and marginals (middle left) Source and target models (middle right) OT matrix on empirical joint distributions and with JDOT proxy joint distribution (right) estimated prediction function $f$.

to illustrate that the OT matrix $\boldsymbol{\gamma}$ obtained using the true empirical distribution $\mathcal{P}_t$ is very similar to the one obtained with the proxy $\mathcal{P}_t^f$ which leads to a very good model for JDOT.

**Choice of $\alpha$.** This is an important parameter balancing the alignment of feature space and labels. A natural choice of the $\alpha$ parameter is obtained by normalizing the range of values of $\mathrm{d}(\mathbf{x}_i^s, \mathbf{x}_j^t)$ with $\alpha = 1/\max_{i,j} \mathrm{d}(\mathbf{x}_i^s, \mathbf{x}_j^t)$. In the numerical experiment section, we show that this setting is very good in two out of three experiments. However, in some cases, better performances are obtained with a cross-validation of this parameter. Also note that $\alpha$ is strongly linked to the smoothness of the loss $\mathcal{L}$ and of the optimal labelling functions and can be seen as a Lipschitz constant in the bound of Theorem 3.1.

**Relation to other optimal transport based DA methods.** Previous DA methods based on optimal transport [14, 17] do not not only differ by the nature of the considered distributions, but also in the way the optimal plan is used to find $f$. They learn a complex mapping between the source and target distributions when the objective is only to estimate a prediction function $f$ on target. To do so, they rely on a barycentric mapping that minimizes only approximately the Wasserstein distance between the distributions. As discussed in Section 4, JDOT uses the optimal plan to propagate and fuse the labels from the source to target. Not only are the performances enhanced, but we also show how this approach is more theoretically well grounded in next section 3.

**Relation to Transport $L^p$ distances.** Recently, Thorpe and co-authors introduced the Transportation $L^p$ distance [20]. Their objective is to compute a meaningful distance between multi-dimensional signals. Interestingly their distance can be seen as optimal transport between two distributions of the form (4) where the functions are known and the label loss $\mathcal{L}$ is chosen as a $L^p$ distance. While their approach is inspirational, JDOT is different both in its formulation, where we introduce a more general class of loss $\mathcal{L}$, and in its objective, as our goal is to estimate the target function $f$ which is not known *a priori*. Finally we show theoretically and empirically that our formulation addresses successfully the problem of domain adaptation.

## 3  A Bound on the Target Error

Let $f$ be an hypothesis function from a given class of hypothesis $\mathcal{H}$. We define the expected loss in the target domain $err_T(f)$ as $err_T(f) \overset{\text{def}}{=} \mathbb{E}_{(\mathbf{x}, y) \sim \mathcal{P}_t} \mathcal{L}(y, f(\mathbf{x}))$. We define similarly $err_S(f)$ for the source domain. We assume the loss function $\mathcal{L}$ to be bounded, symmetric, $k$-lipschitz and satisfying the triangle inequality.

To provide some guarantees on our method, we consider an adaptation of the notion probabilistic Lipschitzness introduced in [21, 22] which assumes that two close instances must have the same labels with high probability. It corresponds to a relaxation of the classic Lipschitzness allowing one to model the marginal-label relatedness such as in Nearest-Neighbor classification, linear classification or cluster assumption. We propose an extension of this notion in a domain adaptation context by assuming that a labeling function must comply with two close instances of each domain w.r.t. a coupling $\Pi$.

**Definition** (**Probabilistic Transfer Lipschitzness**) Let $\mu_s$ and $\mu_t$ be respectively the source and target distributions. Let $\phi : \mathbb{R} \to [0, 1]$. A labeling function $f : \Omega \to \mathbb{R}$ and a joint distribution $\Pi(\mu_s, \mu_t)$ over $\mu_s$ and $\mu_t$ are $\phi$-Lipschitz transferable if for all $\lambda > 0$:

$$Pr_{(\mathbf{x}_1, \mathbf{x}_2) \sim \Pi(\mu_s, \mu_t)} \left[ |f(\mathbf{x}_1) - f(\mathbf{x}_2)| > \lambda d(\mathbf{x}_1, \mathbf{x}_2) \right] \le \phi(\lambda).$$

Intuitively, given a deterministic labeling functions $f$ and a coupling $\Pi$, it bounds the probability of finding pairs of source-target instances labelled differently in a $(1/\lambda)$-ball with respect to $\Pi$.

We can now give our main result (simplified version):

**Theorem 3.1** *Let $f$ be any labeling function of $\in \mathcal{H}$. Let $\Pi^* = \operatorname{argmin}_{\Pi \in \Pi(\mathcal{P}_s, \mathcal{P}_t^f)} \int_{(\Omega \times \mathcal{C})^2} \alpha d(\mathbf{x}_s, \mathbf{x}_t) + \mathcal{L}(y_s, y_t) d\Pi(\mathbf{x}_s, y_s; \mathbf{x}_t, y_t)$ and $W_1(\hat{\mathcal{P}}_s, \hat{\mathcal{P}}_t^f)$ the associated 1-Wasserstein distance. Let $f^* \in \mathcal{H}$ be a Lipschitz labeling function that verifies the $\phi$-probabilistic transfer Lipschitzness (PTL) assumption w.r.t. $\Pi^*$ and that minimizes the joint error $err_S(f^*) + err_T(f^*)$ w.r.t all PTL functions compatible with $\Pi^*$. We assume the input instances are bounded s.t. $|f^*(\mathbf{x}_1) - f^*(\mathbf{x}_2)| \le M$ for all $\mathbf{x}_1, \mathbf{x}_2$. Let $\mathcal{L}$ be any symmetric loss function, $k$-Lipschitz and satisfying the triangle inequality. Consider a sample of $N_s$ labeled source instances drawn from $\mathcal{P}_s$ and $N_t$ unlabeled instances drawn from $\mu_t$, and then for all $\lambda > 0$, with $\alpha = k\lambda$, we have with probability at least $1 - \delta$ that:*

$$err_T(f) \le W_1(\hat{\mathcal{P}}_s, \hat{\mathcal{P}}_t^f) + \sqrt{\frac{2}{c'} \log(\frac{2}{\delta})} \left( \frac{1}{\sqrt{N_S}} + \frac{1}{\sqrt{N_T}} \right) + err_S(f^*) + err_T(f^*) + kM\phi(\lambda).$$

The detailed proof of Theorem 3.1 is given in the supplementary material. The previous bound on the target error above is interesting to interpret. The first two terms correspond to the objective function (5) we propose to minimize accompanied with a sampling bound. The last term $\phi(\lambda)$ assesses the probability under which the probabilistic Lipschitzness does not hold. The remaining two terms involving $f^*$ correspond to the joint error minimizer illustrating that domain adaptation can work only if we can predict well in both domains, similarly to existing results in the literature [23, 24]. If the last terms are small enough, adaptation is possible if we are able to align well $\mathcal{P}_s$ and $\mathcal{P}_t^f$, provided that $f^*$ and $\Pi^*$ verify the PTL. Finally, note that $\alpha = k\lambda$ and tuning this parameter is thus actually related to finding the Lipschitz constants of the problem.

## 4 Learning with Joint Distribution OT

In this section, we provide some details about the JDOT's optimization problem given in Equation (5) and discuss algorithms for its resolution. We will assume that the function space $\mathcal{H}$ to which $f$ belongs is either a RKHS or a function space parametrized by some parameters $\mathbf{w} \in \mathbb{R}^p$. This framework encompasses linear models, neural networks, and kernel methods. Accordingly, we are going to define a regularization term $\Omega(f)$ on $f$. Depending on how $\mathcal{H}$ is defined, $\Omega(f)$ is either a non-decreasing function of the squared-norm induced by the RKHS (so that the representer theorem is applicable) or a squared-norm on the vector parameter. We will further assume that $\Omega(f)$ is continuously differentiable. As discussed above, $f$ is to be learned according to the following optimization problem

$$\min_{f \in \mathcal{H}, \boldsymbol{\gamma} \in \Delta} \sum_{i,j} \boldsymbol{\gamma}_{i,j} \left( \alpha d(\mathbf{x}_i^s, \mathbf{x}_j^t) + \mathcal{L}(y_i^s, f(\mathbf{x}_j^t)) \right) + \lambda \Omega(f) \qquad (6)$$

where the loss function $\mathcal{L}$ is continuous and differentiable with respects to its second variable. Note that while the above problem does not involve any regularization term on the coupling matrix $\boldsymbol{\gamma}$, it is essentially for the sake of simplicity and readability. Regularizers like entropic regularization [25], which is relevant when the number of samples is very large, can still be used without significant change to the algorithmic framework.

**Optimization procedure.** According to the above hypotheses on $f$ and $\mathcal{L}$, Problem (6) is smooth and the constraints are separable according to $f$ and $\boldsymbol{\gamma}$. Hence, a natural way to solve the problem (6) is to rely on alternate optimization w.r.t. both parameters $\boldsymbol{\gamma}$ and $f$. This algorithm well-known as Block Coordinate Descent (BCD) or Gauss-Seidel method (the pseudo code of the algorithm is given in appendix). Block optimization steps are discussed with further details in the following.

Solving with fixed $f$ boils down to a classical OT problem with a loss matrix $\mathbf{C}$ such that $C_{i,j} = \alpha d(\mathbf{x}_i^s, \mathbf{x}_j^t) + \mathcal{L}(y_i^s, f(\mathbf{x}_j^t))$. We can use classical OT solvers such as the network simplex algorithm, but other strategies can be considered, such as regularized OT [25] or stochastic versions [26].

The optimization problem with fixed $\gamma$ leads to a new learning problem expressed as

$$\min_{f \in \mathcal{H}} \quad \sum_{i,j} \gamma_{i,j} \mathcal{L}(y_i^s, f(\mathbf{x}_j^t)) + \lambda \Omega(f) \tag{7}$$

Note how the data fitting term elegantly and naturally encodes the transfer of source labels $y_i^s$ through estimated labels of test samples with a weighting depending on the optimal transport matrix. However, this comes at the price of having a quadratic number $N_s N_t$ of terms, which can be considered as computationally expensive. We will see in the sequel that we can benefit from the structure of the chosen loss to greatly reduce its complexity. In addition, we emphasize that when $\mathcal{H}$ is a RKHS, owing to kernel trick and the representer theorem, problem (7) can be re-expressed as an optimization problem with $N_t$ number of parameters all belonging to $\mathbb{R}$.

Let us now discuss briefly the convergence of the proposed algorithm. Owing to the 2-block coordinate descent structure, to the differentiability of the objective function in Problem (6) and constraints on $f$ (or its kernel trick parameters) and $\gamma$ are closed, non-empty and convex, convergence result of Grippo et al. [27] on 2-block Gauss-Seidel methods directly applies. It states that if the sequence $\{\gamma^k, f^k\}$ produced by the algorithm has limit points then every limit point of the sequence is a critical point of Problem (6).

**Estimating $f$ for least square regression problems.** We detail the use of JDOT for transfer least-square regression problem i.e when $\mathcal{L}$ is the squared-loss. In this context, when the optimal transport matrix $\gamma$ is fixed the learning problem boils down to

$$\min_{f \in \mathcal{H}} \quad \sum_j \frac{1}{n_t} \|\hat{y}_j - f(\mathbf{x}_j^t)\|^2 + \lambda \|f\|^2 \tag{8}$$

where the $\hat{y}_j = n_t \sum_j \gamma_{i,j} y_i^s$ is a weighted average of the source target values. Note that this simplification results from the properties of the quadratic loss and that it may not occur for more complex regression loss.

**Estimating $f$ for hinge loss classification problems.** We now aim at estimating a multiclass classifier with a one-against-all strategy. We suppose that the data fitting is the binary squared hinge loss of the form $\mathcal{L}(y, f(\mathbf{x})) = \max(0, 1 - yf(\mathbf{x}))^2$. In a One-Against-All strategy we often use the binary matrices $\mathbf{P}$ such that $P_{i,k}^s = 1$ if sample $i$ is of class $k$ else $P_{i,k}^s = 0$. Denote as $f_k \in \mathcal{H}$ the decision function related to the $k$-vs-all problem. The learning problem (7) can now be expressed as

$$\min_{f_k \in \mathcal{H}} \quad \sum_{j,k} \hat{P}_{j,k} \mathcal{L}(1, f_k(\mathbf{x}_j^t)) + (1 - \hat{P}_{j,k}) \mathcal{L}(-1, f_k(\mathbf{x}_j^t)) + \lambda \sum_k \|f_k\|^2 \tag{9}$$

where $\hat{\mathbf{P}}$ is the transported class proportion matrix $\hat{\mathbf{P}} = \frac{1}{N_t} \gamma^\top \mathbf{P}^s$. Interestingly this formulation illustrates that for each target sample, the data fitting term is a convex sum of hinge loss for a negative and positive label with weights in $\gamma$.

## 5   Numerical experiments

In this section we evaluate the performance of our method (**JDOT**) on two different transfer tasks of classification and regression on real datasets [2].

**Caltech-Office classification dataset.** This dataset [28] is dedicated to visual adaptation. It contains images from four different domains: *Amazon*, the *Caltech-256* image collection, *Webcam* and *DSLR*. Several features, such as presence/absence of background, lightning conditions, image quality, etc.) induce a distribution shift between the domains, and it is therefore relevant to consider a domain adaptation task to perform the classification. Following [14], we choose deep learning features to represent the images, extracted as the weights of the fully connected 6th layer of the DECAF convolutional neural network [29], pre-trained on ImageNet. The final feature vector is a sparse 4096 dimensional vector.

Table 1: Accuracy on the Caltech-Office Dataset. Best value in bold.

| Domains | Base | SurK | SA | ARTL | OT-IT | OT-MM | JDOT |
|---|---|---|---|---|---|---|---|
| caltech→amazon | 92.07 | 91.65 | 90.50 | 92.17 | 89.98 | **92.59** | 91.54 |
| caltech→webcam | 76.27 | 77.97 | 81.02 | 80.00 | 80.34 | 78.98 | **88.81** |
| caltech→dslr | 84.08 | 82.80 | 85.99 | 88.54 | 78.34 | 76.43 | **89.81** |
| amazon→caltech | 84.77 | 84.95 | 85.13 | 85.04 | 85.93 | **87.36** | 85.22 |
| amazon→webcam | 79.32 | 81.36 | **85.42** | 79.32 | 74.24 | 85.08 | 84.75 |
| amazon→dslr | 86.62 | 87.26 | **89.17** | 85.99 | 77.71 | 79.62 | 87.90 |
| webcam→caltech | 71.77 | 71.86 | 75.78 | 72.75 | **84.06** | 82.99 | 82.64 |
| webcam→amazon | 79.44 | 78.18 | 81.42 | 79.85 | 89.56 | 90.50 | **90.71** |
| webcam→dslr | 96.18 | 95.54 | 94.90 | **100.00** | 99.36 | 99.36 | 98.09 |
| dslr→caltech | 77.03 | 76.94 | 81.75 | 78.45 | **85.57** | 83.35 | 84.33 |
| dslr→amazon | 83.19 | 82.15 | 83.19 | 83.82 | **90.50** | **90.50** | 88.10 |
| dslr→webcam | 96.27 | 92.88 | 88.47 | **98.98** | 96.61 | 96.61 | 96.61 |
| **Mean** | 83.92 | 83.63 | 85.23 | 85.41 | 86.02 | 86.95 | **89.04** |
| **Mean rank** | 5.33 | 5.58 | 4.00 | 3.75 | 3.50 | 2.83 | 2.50 |
| **p-value** | $< 0.01$ | $< 0.01$ | 0.01 | 0.04 | 0.25 | 0.86 | – |

Table 2: Accuracy on the Amazon review experiment. Maximum value in bold font.

| Domains | NN | DANN | JDOT (mse) | JDOT (Hinge) |
|---|---|---|---|---|
| books→dvd | 0.805 | **0.806** | 0.794 | 0.795 |
| books→kitchen | 0.768 | 0.767 | 0.791 | **0.794** |
| books→electronics | 0.746 | 0.747 | 0.778 | **0.781** |
| dvd→books | 0.725 | 0.747 | 0.761 | **0.763** |
| dvd→kitchen | 0.760 | 0.765 | 0.811 | **0.821** |
| dvd→electronics | 0.732 | 0.738 | 0.778 | **0.788** |
| kitchen→books | 0.704 | 0.718 | **0.732** | 0.728 |
| kitchen→dvd | 0.723 | 0.730 | 0.764 | **0.765** |
| kitchen→electronics | **0.847** | 0.846 | 0.844 | 0.845 |
| electronics→books | 0.713 | 0.718 | 0.740 | **0.749** |
| electronics→dvd | 0.726 | 0.726 | **0.738** | 0.737 |
| electronics→kitchen | 0.855 | 0.850 | 0.868 | **0.872** |
| **Mean** | 0.759 | 0.763 | 0.783 | **0.787** |
| **p-value** | 0.004 | 0.006 | 0.025 | – |

We compare our method with four other methods: the surrogate kernel approach ([4], denoted **SurK**), subspace adaptation for its simplicity and good performances on visual adaptation ([8], **SA**), Adaptation Regularization based Transfer Learning ([30], **ARTL**), and the two variants of regularized optimal transport [14]: entropy-regularized **OT-IT** and classwise regularization implemented with the Majoration-Minimization algorithm **OT-MM**, that showed to give better results in practice than its group-lasso counterpart. The classification is conducted with a SVM together with a linear kernel for every method. Its results when learned on the source domain and tested on the target domain are also reported to serve as baseline (**Base**). All the methods have hyper-parameters, that are selected using the reverse cross-validation of Zhong and colleagues [31].The dimension d for **SA** is chosen from $\{1, 4, 7, \ldots, 31\}$. The entropy regularization for **OT-IT** and **OT-MM** is taken from $\{10^2, \ldots, 10^5\}$, $10^2$ being the minimum value for the Sinkhorn algorithm to prevent numerical errors. Finally the $\eta$ parameter of **OT-MM** is selected from $\{1, \ldots, 10^5\}$ and the $\alpha$ in **JDOT** from $\{10^{-5}, 10^{-4}, \ldots, 1\}$.

The classification accuracy for all the methods is reported in Table 1. We can see that **JDOT** is consistently outperforming the baseline (5 points in average), indicating that the adaptation is successful in every cases. Its mean accuracy is the best as well as its average ranking. We conducted a Wilcoxon signed-rank test to test if **JDOT** was statistically better than the other methods, and report the p-value in the tables. This test shows that **JDOT** is statistically better than the considered methods, except for OT based ones that where state of the art on this dataset [14].

**Amazon review classification dataset** We now consider the *Amazon review dataset* [32] which contains online reviews of different products collected on the Amazon website. Reviews are encoded with bag-of-word unigram and bigram features as input. The problem is to predict positive (higher than 3 stars) or negative (3 stars or less) notation of reviews (binary classification). Since different

Table 3: Comparison of different methods on the Wifi localization dataset. Maximum value in bold.

| Domains | KRR | SurK | DIP | DIP-CC | GeTarS | CTC | CTC-TIP | JDOT |
|---|---|---|---|---|---|---|---|---|
| t1 → t2 | 80.84±1.14 | 90.36±1.22 | 87.98±2.33 | 91.30±3.24 | 86.76 ± 1.91 | 89.36±1.78 | 89.22±1.66 | **93.03 ± 1.24** |
| t1 → t3 | 76.44±2.66 | **94.97±1.29** | 84.20±4.29 | 84.32±4.57 | 90.62±2.25 | 94.80±0.87 | 92.60 ± 4.50 | 90.06 ± 2.01 |
| t2 → t3 | 67.12±1.28 | 85.83 ± 1.31 | 80.58 ± 2.10 | 81.22 ± 4.31 | 82.68 ± 3.71 | 87.92 ± 1.87 | **89.52 ± 1.14** | 86.76 ± 1.72 |
| hallway1 | 60.02 ±2.60 | 76.36 ± 2.44 | 77.48 ± 2.68 | 76.24± 5.14 | 84.38 ± 1.98 | 86.98 ± 2.02 | 86.78 ± 2.31 | **98.83±0.58** |
| hallway2 | 49.38 ± 2.30 | 64.69 ±0.77 | 78.54 ± 1.66 | 77.8± 2.70 | 77.38 ± 2.09 | 87.74 ± 1.89 | 87.94 ± 2.07 | **98.45±0.67** |
| hallway3 | 48.42 ±1.32 | 65.73 ± 1.57 | 75.10± 3.39 | 73.40± 4.06 | 80.64 ± 1.76 | 82.02± 2.34 | 81.72 ± 2.25 | **99.27±0.41** |

words are employed to qualify the different categories of products, a domain adaptation task can be formulated if one wants to predict positive reviews of a product from labelled reviews of a different product. Following [33, 11], we consider only a subset of four different types of product: books, DVDs, electronics and kitchens. This yields 12 possible adaptation tasks. Each domain contains 2000 labelled samples and approximately 4000 unlabelled ones. We therefore use these unlabelled samples to perform the transfer, and test on the 2000 labelled data.

The goal of this experiment is to compare to the state-of-the-art method on this subset, namely Domain adversarial neural network ([11], denoted **DANN**), and to show the versatility of our method that can adapt to any type of classifier. The neural network used for all methods in this experiment is a simple 2-layer model with sigmoid activation function in the hidden layer to promote non-linearity. 50 neurons are used in this hidden layer. For **DANN**, hyper-parameters are set through the reverse cross-validation proposed in [11], and following the recommendation of authors the learning rate is set to $10^{-3}$. In the case of **JDOT**, we used the heuristic setting of $\alpha = 1/\max_{i,j} d(\mathbf{x}_i^s, \mathbf{x}_j^t)$, and as such we do not need any cross-validation. The squared Euclidean norm is used for both metric in feature space and we test as loss functions both mean squared errors (mse) and Hinge losses. 10 iterations of the block coordinate descent are realized. For each method, we stop the learning process of the network after 5 epochs. Classification accuracies are presented in table 2. The neural network (**NN**), trained on source and tested on target, is also presented as a baseline. **JDOT** surpasses **DANN** in 11 out of 12 tasks (except on books→dvd). The Hinge loss is better in than mse in 10 out of 12 cases, which is expected given the superiority of the Hinge loss on classification tasks [19].

**Wifi localization regression dataset** For the regression task, we use the cross-domain indoor Wifi localization dataset that was proposed by Zhang and co-authors [4], and recently studied in [5]. From a multi-dimensional signal (collection of signal strength perceived from several access points), the goal is to locate the device in a hallway, discretized into a grid of 119 squares, by learning a mapping from the signal to the grid element. This translates as a regression problem. As the signals were acquired at different time periods by different devices, a shift can be encountered and calls for an adaptation. In the remaining, we follow the exact same experimental protocol as in [4, 5] for ease of comparison. Two cases of adaptation are considered: **transfer across periods**, for which three time periods t1, t2 and t3 are considered, and **transfer across devices**, where three different devices are used to collect the signals in the same straight-line hallways (hallway1-3), leading to three different adaptation tasks in both cases.

We compare the result of our method with several state-of-the-art methods: kernel ridge regression with RBF kernel (**KRR**), surrogate kernel ([4], denoted **SurK**), domain-invariant projection and its cluster regularized version ([7], denoted respectively **DIP** and **DIP-CC**), generalized target shift ([34], denoted **GeTarS**), and conditional transferable components, with its target information preservation regularization ([5], denoted respectively **CTC** and **CTC-TIP**). As in [4, 5], the hyper-parameters of the competing methods are cross-validated on a small subset of the target domain. In the case of **JDOT**, we simply set the $\alpha$ to the heuristic value of $\alpha = 1/\max_{i,j} d(\mathbf{x}_i^s, \mathbf{x}_j^t)$ as discussed previously, and $f$ is estimated with kernel ridge regression.

Following [4], the accuracy is measured in the following way: the prediction is said to be correct if it falls within a range of three meters in the transfer across periods, and six meters in the transfer across devices. For each experiment, we randomly sample sixty percent of the source and target domain, and report the mean and standard deviation of ten repetitions accuracies in Table 3. For transfer across periods, **JDOT** performs best in one out of three tasks. For transfer across devices, the superiority of **JDOT** is clearly assessed, for it reaches an average score $> 98\%$, which is at least ten points ahead of the best competing method for every task. Those extremely good results could be explained by the fact that using optimal transport allows to consider large shifts of distribution, for which divergences (such as maximum mean discrepancy used in **CTC**) or reweighting strategies can not cope with.

# 6 Discussion and conclusion

We have presented in this paper the Joint Distribution Optimal Transport for domain adaptation, which is a principled way of performing domain adaptation with optimal transport. JDOT assumes the existence of a transfer map that transforms a source domain joint distribution $\mathcal{P}_s(X, Y)$ into a target domain equivalent version $\mathcal{P}_t(X, Y)$. Through this transformation, the alignment of both feature space and conditional distributions is operated, allowing to devise an efficient algorithm that simultaneously optimizes for a coupling between $\mathcal{P}_s$ and $\mathcal{P}_t$ and a prediction function that solves the transfer problem. We also proved that learning with **JDOT** is equivalent to minimizing a bound on the target distribution. We have demonstrated through experiments on classical real-world benchmark datasets the superiority of our approach w.r.t. several state-of-the-art methods, including previous work on optimal transport based domain adaptation, domain adversarial neural networks or transfer components, on a variety of task including classification and regression. We have also showed the versatility of our method, that can accommodate with several types of loss functions (mse, hinge) or class of hypothesis (including kernel machines or neural networks). Potential follow-ups of this work include a semi-supervised extension (using unlabelled examples in source domain) and investigating stochastic techniques for solving efficiently the adaptation. From a theoretical standpoint, future works include a deeper study of probabilistic transfer lipschitzness and the development of guarantees able to take into the complexity of the hypothesis class and the space of possible transport plans.

## Acknowledgements

This work benefited from the support of the project OATMIL ANR-17-CE23-0012 of the French National Research Agency (ANR), the Normandie Projet GRR-DAISI, European funding FEDER DAISI and CNRS funding from the Défi Imag'In. The authors also wish to thank Kai Zhang and Qiaojun Wang for providing the Wifi localization dataset.

## Footnotes

[2]Open Source Python implementation of JDOT: `https://github.com/rflamary/JDOT`

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
