[Supplementary Material]

# Joint distribution optimal transportation for domain adaptation
## Supplementary Material

## 1 Illustration on a simple example

We illustrate the behavior of our method on a 3-class toy example (Figure 1). We consider a classification problem using the hinge loss and $\mathcal{H}$ is a Reproducing Kernel Hilbert Space. Source domain samples are drawn from three different 2D Gaussian distributions with with different centers and standard deviations. The target domain is obtained rotating the source distribution by $\pi/4$ radian. Two types of kernel are considered: linear and RBF. In Figure 1.a, one can observe on the first column of images that using directly a classifier learned on the source domain leads to bad performances because of the rotation. We then show the iterations of the block coordinate descent which allows one to recover the true labels of the target domain. It is also interesting to examine the impact of the $\alpha$ parameter on the success of the method. In Figure 1.b, we show the evolution of classification accuracy for six different $\alpha$ in the case of RBF kernel. Relying mostly on the label cost ($\alpha = \{0.1\}$) leads to a deterioration of the final accuracy. Using only the input space distance ($\alpha = \{50, 100\}$), which is equivalent to [2], allows a performance gain. But it is clear that using both losses with $\alpha = \{0.5, 1, 10\}$ leads to the best performance. Also note the small number of iterations required ($< 10$) for achieving a steady state.

Figure 1: **Illustration on a toy example**. (a): Decision boundaries for linear and RBF kernels on selected iterations. The source domain is depicted with crosses, while the target domain samples are class-colored circles. (b): Evolution of the accuracy along 15 iterations of the method for different values of the $\alpha$ parameter;

| Iter | caltech → amazon | dslr → amazon | webcam → caltech |
|------|------------------|---------------|------------------|
| 0 | 89.14 | 80.9 | 75.6 |
| 1 | 91.75 | 86.22 | 80.23 |
| 2 | 91.44 | 86.95 | 81.75 |
| 3 | 91.54 | 87.68 | 82.64 |
| 4 | 91.44 | 87.68 | 83.26 |
| 5 | 91.65 | 88.0 | 83.35 |
| 6 | 91.86 | 88.1 | 83.17 |
| 7 | 92.17 | 87.89 | 83.08 |
| 8 | 92.28 | 87.58 | 83.26 |
| 9 | 92.28 | 87.58 | 83.26 |
| 10 | 92.28 | 87.68 | 83.35 |
| 11 | 92.28 | 87.79 | 83.44 |
| 12 | 92.28 | 87.79 | 83.44 |
| 13 | 92.28 | 87.79 | 83.44 |
| 14 | 92.28 | 87.79 | 83.44 |

Table 1: Accuracy of the estimated model along BCD iterations on Caltech-office dataset

## 2 Block coordinate descent algorithm for solving JDOT

We give in algorithm 1 an overview of the block coordinate descent algorithm used for solving JDOT.

---
**Algorithm 1** Optimization with Block Coordinate Descent
---
Initialize function $f^0$ and set $k = 1$
Set $\alpha$ and $\lambda$
**while** not converged **do**
    $\gamma^k \leftarrow$ Solve OT problem (3 in paper) with fixed $f^{k-1}$
    $f^k \leftarrow$ Solve learning problem (7 in paper) with fixed $\gamma^k$
    $k \leftarrow k + 1$
**end while**
---

## 3 BCD iterations on real data

We report in Table 1, for a fixed set of parameter (no CV), the evolution of the empirical error along the iterations of the 15 first iterations of the BCD on a real dataset. We can see that generally the result stabilizes at around 10 iterations. We can also observe that the increase in performance is not monotonic, contrary to the toy example.

## 4 Proof of Theorem 3.1

We first recall some hypothesis used for this theorem.

$\mathcal{H} \subset \mathcal{C}^\Omega$ is the hypothesis class. $\mathcal{L} : \mathcal{C} \times \mathcal{C} \rightarrow \mathbb{R}^+$ is the loss function measuring the discrepancy between two labels. This loss is assumed to be symmetric, bounded and

$k$-lipschitz in its second argument, *i.e.* there exists $k$ such that for any $y_1, y_2, y_3 \in \mathcal{C}$:

$$|\mathcal{L}(y_1, y_2) - \mathcal{L}(y_1, y_3)| \leq k|y_2 - y_3|.$$

$\mathcal{P}_t$ and $\mathcal{P}_s$ are respectively the target and source distributions over $\Omega \times \mathcal{C}$, with $\mu_t$ and $\mu_s$ the respective marginals over $\Omega$. The expected loss in the target domain $err_T(f)$ is defined for any $f \in \mathcal{H}$ as

$$err_T(f) \stackrel{\text{def}}{=} \mathop{\mathbb{E}}_{(\mathbf{x}, y) \sim \mathcal{P}_t} \mathcal{L}(y, f(\mathbf{x})).$$

We can similarly define $err_S(f)$ in the source domain and the expected inter function loss $err_T(f, g) = \mathbb{E}_{(\mathbf{x}, y) \sim \mathcal{P}_t} \mathcal{L}(g(\mathbf{x}), f(\mathbf{x}))$.

The proxy $\mathcal{P}_t^f$ over $\Omega \times \mathcal{C}$ of $\mathcal{P}_t$ w.r.t. to $\mu_t$ and $f$ is defined as: $\mathcal{P}_t^f = (\mathbf{x}, f(\mathbf{x}))_{\mathbf{x} \sim \mu_t}$.

We consider the following transport loss function:

$$W_1(\mathcal{P}_s, \mathcal{P}_t^f) = \inf_{\Pi \in \Pi(\mathcal{P}_s, \mathcal{P}_t^f)} \int_{(\Omega \times \mathcal{C})^2} \alpha d(\mathbf{x}_s, \mathbf{x}_t) + \mathcal{L}(y_s, y_t) \mathbf{d}\Pi((\mathbf{x}_s, y_s), (\mathbf{x}_t, y_t)).$$

We now recall the definition of the theorem with all the assumptions.

**Theorem 4.1** *Let $\mathcal{H} \subset \mathcal{C}^\Omega$ be the hypothesis class where $\Omega$ is a compact mesurable space of finite dimension accompanied with a metric $d$, and $\mathcal{C}$ is the output space. Let $f$ be any labeling function of $\in \mathcal{H}$. Let $\mathcal{P}_s$, $\mathcal{P}_t$, $\mathcal{P}_t^f$ be three probability distributions over $\Omega \times \mathcal{C}$ with bounded support, with $\mathcal{P}_t^f$ defined w.r.t. the marginal $\mu_t$ of $\mathcal{P}_t$ and $f$, accompanied with a sample of $N_s$ labeled source instances drawn from $\mathcal{P}_s$ and $N_t$ unlabeled instances drawn from $\mu_t$ and labeled by $f$, such that $\mathcal{P}_s$ and $\mathcal{P}_t^f$ and the associated samples follow the assumptions of Theorem 5.1. Let $\Pi^* = \operatorname{argmin}_{\Pi \in \Pi(\mathcal{P}_s, \mathcal{P}_t^f)} \int_{(\Omega \times \mathcal{C})^2} \alpha d(\mathbf{x}_s, \mathbf{x}_t) + \mathcal{L}(y_s, y_t) \mathbf{d}\Pi((\mathbf{x}_s, y_s), (\mathbf{x}_t, y_t))$. Let $f^*$ be a Lipschitz labeling function of $\mathcal{H}$, that verifies the $\phi$-probabilistic transfer Lipschitzness (PTL) assumption with respect to $\Pi^*$ and that minimizes the joint error $err_S(f^*) + err_T(f^*)$ w.r.t all compatible PTL functions with $\Pi^*$. We assume the instance space $\mathcal{X} \subseteq \Omega$ is bounded[1] such that $|f^*(\mathbf{x}_1) - f^*(\mathbf{x}_2)| \leq M$ for all $\mathbf{x}_1, \mathbf{x}_2 \in \mathcal{X}^2$. Let $\mathcal{L}$ be any loss function symmetric, $k$-lipschitz and that satisfies the triangle inequality. Then, there exists, $c'$ and $N$, such that for $N_s > N$ and $N_t > N$, for all $\lambda > 0$, with $\alpha = k\lambda$, we have with probability at least $1 - \delta$:*

$$err_T(f) \leq W_1(\hat{\mathcal{P}}_s, \hat{\mathcal{P}}_t^f) + \sqrt{\frac{2}{c'} \log(\frac{2}{\delta})} \left( \frac{1}{\sqrt{N_s}} + \frac{1}{\sqrt{N_t}} \right)$$
$$+ err_S(f^*) + err_T(f^*) + kM\phi(\lambda).$$

**Proof**

$$
\begin{aligned}
\text{err}_T(f) \quad &= E_{(\mathbf{x}, y) \sim P_t} \mathcal{L}(y, f(\mathbf{x})) \\
&\leq E_{(\mathbf{x}, y) \sim P_t} \mathcal{L}(y, f^*(\mathbf{x})) + \mathcal{L}(f^*(\mathbf{x}), f(\mathbf{x})) \\
&= E_{(\mathbf{x}, y) \sim P_t} \mathcal{L}(f(\mathbf{x}), f^*(\mathbf{x})) + err_T(f^*) && (1) \\
&= E_{(\mathbf{x}, y) \sim \mathcal{P}_t^f} \mathcal{L}(f(\mathbf{x}), f^*(\mathbf{x})) + err_T(f^*) && (2) \\
&= err_{Tf}(f^*) - err_S(f^*) + err_S(f^*) + err_T(f^*) \\
&\leq |err_{Tf}(f^*) - err_S(f^*)| + err_S(f^*) + err_T(f^*) && (3)
\end{aligned}
$$

Line (1) is due to the symmetry of the loss. Line (2) comes from the fact that:
$$E_{(\mathbf{x},y)\sim\mathcal{P}_t}L(f(\mathbf{x}),f^*(\mathbf{x})) = E_{(\mathbf{x},f(\mathbf{x}))\sim\mathcal{P}_t^f}L(f(\mathbf{x}),f^*(\mathbf{x})) \stackrel{\text{def}}{=} err_{Tf}(f^*(\mathbf{x})).$$

Now, we have

$$
\begin{aligned}
&|err_{Tf}(f^*) - err_S(f^*)| \\
&= \left| \iint_{\Omega\times\mathcal{C}} \mathcal{L}(y,f^*(\mathbf{x}))(\mathcal{P}_t^f(\boldsymbol{X}=\mathbf{x},Y=y) - \mathcal{P}_s(\boldsymbol{X}=\mathbf{x},Y=y))d\mathbf{x}dy \right| \\
&= \left| \int_{\Omega\times\mathcal{C}} \mathcal{L}(y,f^*(\mathbf{x}))d(\mathcal{P}_t^f - \mathcal{P}_s) \right| \\
&\leq \int_{(\Omega\times\mathcal{C})^2} \left| \mathcal{L}(y_t^f,f^*(\mathbf{x}_t)) - \mathcal{L}(y_s,f^*(\mathbf{x}_s)) \right| d\Pi^*((\mathbf{x}_s,y_s),(\mathbf{x}_t,y_t^f)) \qquad (4) \\
&= \int_{(\Omega\times\mathcal{C})^2} \Big| \mathcal{L}(y_t^f,f^*(\mathbf{x}_t)) - \mathcal{L}(y_t^f,f^*(\mathbf{x}_s)) + \\
&\qquad\qquad \mathcal{L}(y_t^f,f^*(\mathbf{x}_s)) - \mathcal{L}(y_s,f^*(\mathbf{x}_s)) \Big| d\Pi^*((\mathbf{x}_s,y_s),(\mathbf{x}_t,y_t^f)) \\
&\leq \int_{(\Omega\times\mathcal{C})^2} \Big| \mathcal{L}(y_t^f,f^*(\mathbf{x}_t)) - \mathcal{L}(y_t^f,f^*(\mathbf{x}_s)) \Big| \\
&\qquad\qquad + \Big| \mathcal{L}(y_t^f,f^*(\mathbf{x}_s)) - \mathcal{L}(y_s,f^*(\mathbf{x}_s)) \Big| d\Pi^*((\mathbf{x}_s,y_s),(\mathbf{x}_t,y_t^f)) \\
&\leq \int_{(\Omega\times\mathcal{C})^2} k \left| f^*(\mathbf{x}_t) - f^*(\mathbf{x}_s) \right| + \\
&\qquad\qquad \Big| \mathcal{L}(y_t^f,f^*(\mathbf{x}_s)) - \mathcal{L}(y_s,f^*(\mathbf{x}_s)) \Big| d\Pi^*((\mathbf{x}_s,y_s),(\mathbf{x}_t,y_t^f)) \qquad (5) \\
&\leq k*M*\phi(\lambda) + \int_{(\Omega\times\mathcal{C})^2} k\lambda d(\mathbf{x}_t,\mathbf{x}_s) + \\
&\qquad\qquad \Big| \mathcal{L}(y_t^f,f^*(\mathbf{x}_s)) - \mathcal{L}(y_s,f^*(\mathbf{x}_s)) \Big| d\Pi^*((\mathbf{x}_s,y_s),(\mathbf{x}_t,y_t^f)) \qquad (6) \\
&\leq \int_{(\Omega\times\mathcal{C})^2} \alpha d(\mathbf{x}_s,\mathbf{x}_t) + \mathcal{L}(y_t^f,y_s)d\Pi^*((\mathbf{x}_s,y_s),(\mathbf{x}_t,y_t^f)) + k*M*\phi(\lambda) \quad(7) \\
&\leq \int_{(\Omega\times\mathcal{C})^2} \alpha d(\mathbf{x}_s,\mathbf{x}_t) + \mathcal{L}(y_s,y_t^f)d\Pi^*((\mathbf{x}_s,y_s),(\mathbf{x}_t,y_t^f)) + k*M*\phi(\lambda) \quad(8) \\
&= W_1(\mathcal{P}_s,\mathcal{P}_t^f) + k*M*\phi(\lambda). \qquad (9)
\end{aligned}
$$

Line (4) is a consequence of the duality form of the Kantorovitch-Rubinstein theorem saying that for any coupling $\Pi \in \Pi(P_s, P_t^f)$, we have:

$$
\begin{aligned}
&\left| \int_{\Omega\times\mathcal{C}} \mathcal{L}(y,f^*(\mathbf{x}))d(\mathcal{P}_t^f - \mathcal{P}_s) \right| \\
&= \left| \int_{(\Omega\times\mathcal{C})^2} \mathcal{L}(y_t^f,f^*(\mathbf{x}_t)) - \mathcal{L}(y_s,f^*(\mathbf{x}_s))d\Pi((\mathbf{x}_s,y_s),(\mathbf{x}_t,y_t^f)) \right| \\
&\leq \int_{(\Omega\times\mathcal{C})^2} \left| \mathcal{L}(y_t^f,f^*(\mathbf{x}_t)) - \mathcal{L}(y_s,f^*(\mathbf{x}_s)) \right| d\Pi((\mathbf{x}_s,y_s),(\mathbf{x}_t,y_t^f)).
\end{aligned}
$$

Since the inequality is true for any coupling, it is then also true for $\Pi^*$. Inequality (5) is due to the $k$-lipschitzness of the loss $\mathcal{L}$ in its second argument. Inequality (6)

uses the fact that $f^*$ and $\Pi^*$ verify the probabilistic transfer Lipschitzness property with probability $1 - \phi(\lambda)$, additionally, taking into account that the deviation between 2 instances with respect to $f^*$ is bounded by $M$ we have the additional term $kM\phi(\lambda)$ that covers the regions where the PTL does not hold. (7) is obtained by the symmetry of $d$, the use of triangle inequality on $\mathcal{L}$ and by replacing $k\lambda$ by $\alpha$. Other inequalities above are due the use of triangle inequality or properties of the absolute value. The last line (9) is due to the definition of $\Pi^*$.

Now, note that by the use of triangle inequality:

$$W_1(\mathcal{P}_s, \mathcal{P}_t^f) \leq W_1(\mathcal{P}_s, \hat{\mathcal{P}}_s) + W_1(\hat{\mathcal{P}}_s, \hat{\mathcal{P}}_t^f) + W_1(\hat{\mathcal{P}}_t^f, \mathcal{P}_t^f) \tag{10}$$

$$\leq W_1(\hat{\mathcal{P}}_s, \hat{\mathcal{P}}_t^f) + \sqrt{\frac{2}{c'}\log(\frac{2}{\delta})}\left(\frac{1}{\sqrt{N_s}} + \frac{1}{\sqrt{N_t}}\right). \tag{11}$$

Indeed, the cost function $\mathcal{D}((\mathbf{x}_s, y_s), (\mathbf{x}_t, y_t)) = \alpha d(\mathbf{x}_1, \mathbf{x}_2) + \mathcal{L}(y_1, y_2)$ defines a distance over $(\Omega \times \mathcal{L})^2$, assuming that $\mathcal{P}_s$ and $\mathcal{P}_t^f$ have bounded support and the fact that our loss function is bounded, we can apply Theorem 5.1 (presented below) on $W_1(\mathcal{P}_s, \hat{\mathcal{P}}_s)$ and $W_1(\hat{\mathcal{P}}_t^f, \mathcal{P}_t)$ above (with probability $\delta/2$ each). The two settings may have different constants $N$ and $c'$ and and we consider the maximum $N$ and the minimum $c'$ that comply with both cases.

Combining inequalities (3), (9), inequality (11) and the use of the union bound, the theorem holds with probability at least $1 - \delta$ for any $f \in \mathcal{H}$. $\square$

Note that, additionally to the analysis in the paper, a link can be made with classic generalization bounds when the two distributions are equal, *i.e.* $\mathcal{P}_s = \mathcal{P}_t$. Indeed, if we can choose $f^*$ as the true labeling function on source/target domains such that $f^*$ is strongly $\phi$-lipschitz w.r.t. $\Pi^*$ (*i.e.* $\phi(\lambda)$ is almost 0), then the bound is similar to a classic generalization bound: terms involving $f^*$ are null and using the same sample for source and target $d(\mathbf{x}_1, \mathbf{x}_2) = 0$ w.r.t the best alignment. Thus, it remains only the label loss which corresponds to a classic supervised learning loss.

# 5  Empirical concentration result for Wasserstein distance

We give now the result from Bolley and co-authors used in the previous section.

**Theorem 5.1 (from [1], Theorem 1.1.)** *Let $\mu$ be a probability measure in $Z$ so that for some $\alpha > 0$ we have for any $\mathbf{z}'$ $\int_{\mathbb{R}^d} e^{\alpha dist(\mathbf{z}, \mathbf{z}')^2} d\mu < \infty$ and $\hat{\mu} = \frac{1}{N}\sum_{i=1}^N \delta_{z_i}$ be the associated empirical measure defined on a sample of independent variables $\{\mathbf{z}_i\}_{i=1}^N$ drawn from $\mu$. Then, for any $d' > dim(Z)$ and $c' < c$, there exists some constant $N_0$ depending on $d'$ and some square exponential moments of $\mu$ such that for any $\epsilon > 0$ and $N \geq N_0 \max(\epsilon^{-(d'+2)}, 1)$,*

$$P[W_1(\mu, \hat{\mu}) > \epsilon] \leq \exp\left(-\frac{c'}{2}N\epsilon^2\right)$$

*where $c'$ can be calculated explicitly.*

Note that $c$ is such that $\mu$ verifies for any measure $\nu$ the Talagrand (transport) inequality $T_1(c) : W_1(\mu, \nu) \leq \sqrt{\frac{2}{c}H(\nu|\mu)}$ with $H$ is the relative entropy. $T_1(c)$ holds

when for some $\alpha > 0$ and for any $\mathbf{z}'$: $\int_{\mathbb{R}^d} e^{\alpha dist(\mathbf{z},\mathbf{z}')^2} d\mu(\mathbf{z}) < \infty$, and $c$ can be found explicitly [1].

## Footnotes

[1] Since the input space is bounded by say a constant $K$: $\|\mathbf{x}\| \leq K$, since $f^*$ is supposed $l$-Lipschitz, then we have for any $\mathbf{x}_1, \mathbf{x}_2$: $|f(\mathbf{x}_1) - f(\mathbf{x}_2)| \leq l\|\mathbf{x}_1 - \mathbf{x}_2\| \leq 2lK = M$.