[Reviews · NeurIPS 2017]

Reviewer 1



The manuscript introduces a new technique for unsupervised domain adaptation (DA) based on optimal transport (OT). Previous related techniques (that are state-of-the-art) use optimal transport to find a mapping between the marginal distributions of source and target. The manuscript proposes to transports the joint distributions instead of the marginals. As the target does not have labels, the learned prediction function on the target is used as proxy. The consequence is an iterative algorithm that alternates between 1) estimating the transport for fixed data points and labeling, and 2) estimating the labeling function for fixed transport. As such, the algorithm can be seen as a self-learning inspired extension of previous work, which executed 1) and 2) only once. In addition to the algorithm, the manuscript contains a high probability generalization bound, similar to the discrepancy-based bounds of Ben-David et al [24]. It is used as justification that minimizing the joint OT distance should result in better predictions. Formally, this is not valid, however, because the bound is not uniform in the labeling function, so it might not hold for the result of the minimization. There are also experiments on three standard datasets that show that the proposed method achieve state-of-the-art performance (though the differences to the previous OT based methods are small). strength: - the paper is well written, the idea makes sense - the method seems practical, as both repeated steps resemble previous OT-based DA work - the theory seems correct (but is not enough to truly justify the method, see below) - the experiments are done well, including proper model selection (which is rare in DA) - the experimental results are state-of-the-art weaknesses: - the manuscript is mainly a continuation of previous work on OT-based DA - while the derivations are different, the conceptual difference is previous work is limited - theoretical results and derivations are w.r.t. the loss function used for learning (e.g. hinge loss), which is typically just a surrogate, while the real performance measure would be 0/1 loss. This also makes it hard to compare the bounds to previous work that used 0-1 loss - the theorem assumes a form of probabilistic Lipschitzness, which is not explored well. Previous discrepancy-based DA theory does not need Prob.Lipschitzness and is more flexible in this respect. - the proved bound (Theorem 3.1) is not uniform w.r.t. the labeling function $f$. Therefore, it does not suffice as a justification for the proposed minimization procedure. - the experimental results do not show much better results than previous OT-based DA methods - as the proposed method is essentially a repeated application of the previous work, I would have hoped to see real-data experiments exploring this. Currently, performance after different number of alternating steps is reported only in the supplemental material on synthetic data. - the supplemental material feels rushed in some places. E.g. in the proof of Theorem 3.1, the first inequality on page 4 seems incorrect (as the integral is w.r.t. a signed measure, not a prob.distr.). I believe the proof can be fixed, though, because the relation holds without absolute values, and it's not necessary to introduce these in (3) anyway. - In the same proof, Equations (7)/(8) seem identical to (9)/(10) questions to the authors: - please comment if the effect of multiple BCD on real data is similar to the synthetic case *************************** I read the author response and I am still in favor of accepting the work.

Reviewer 2



This paper proposes a straightforward domain adaptation method using the optimal transport plan. Since the optimal transport is coupled with the prediction function of the target domain, it is not immediately clear how the optimal transport can help find a good prediction function for the target domain. Nonetheless, Section 3 presents a theoretical justification based on some assumptions. It looks like the justification is reasonable. The paper provides comparison experiments on three datasets, but doest not have any ablation studies or analyses of the method. Some of the practical choices seem like arbitrary, such as the distance between data points and the loss between labels. What are the effects of different distance/loss metrics over the final results? Are there any good principles to choose between the different metrics? Overall, the proposed method is simple and yet reads reasonable. The experiments lack some points but those are not critical. In the rebuttal, it would be great to see some experimental answers to the above questions.

Reviewer 3



Summary: - The authors propose theory and an algorithm for unsupervised domain adaptation. Unlike much work on this topic, they do not make the covariate shift assumption, but introduce the notion of Probabilistic Transfer Lipschitzness (PTL), an Lipschitz-assumption about the labelling function across domains. They give a bound on the transfer/generalization error in terms of the distance between source and target joint distributions as well as the error of the best PTL hypothesis. An experimental evaluation is performed on three real-world datasets, with the proposed method JDOT performing very well. Clarity: - Overall, the paper is clear and easy to follow. - What is the motivation/implications of choosing an additive/separable loss? It involves a trade-off between marginals yes, but how general is this formulation? - How much does the example in Figure 1 rely on the target function being an offset version of the source function? What would happen if the offset was in the y-direction instead? - In the abstract, the authors state that “Our work makes the following assumption: there exists a non-linear transformation between the joint feature/label space distributions of the two domain Ps and Pt.” Is this meant to describe the PTL assumption? In that case, I think the interpretation made in the abstract should be made in the text. Theory: - I would like the authors to comment on the tightness/looseness of the bound (Thm 3.1). For example, what happens in the limit of infinite samples? Am I right in understanding that the W1 term remains, unless the source and target distributions coincide? If they are different but have common support, and we have access to infinite samples, is the bound loose? - Could the authors comment on the relation between the covariate shift assumption and the proposed PTL assumption? Experiments: - There appears to be a typo in Table 1. Does Tloss refer to JDOT? Tloss is not introduced in the text.